

# The role of CXCL10 as a biomarker for immunological response among patients with leprosy: a systematic literature review

Flora Ramona Sigit Prakoeswa[1,2], Nabila Haningtyas[3],
Listiana Masyita Dewi[4], Ellen Josephine Handoko[3],
Moch. Tabriz Azenta[5] and Muhana Fawwazy Ilyas[6,7]

[1] Department of Dermatology and Venereology, Faculty of Medicine, Muhammadiyah University of Surakarta, Surakarta, Central Java, Indonesia
[2] Department of Dermatology and Venereology, PKU Muhammadiyah Surakarta Hospital, Surakarta, Central Java, Indonesia
[3] Faculty of Medicine, Sebelas Maret University, Surakarta, Central Java, Indonesia
[4] Department of Microbiology, Faculty of Medicine, Muhammadiyah University of Surakarta, Surakarta, Central Java, Indonesia
[5] Faculty of Medicine, Muhammadiyah University of Surakarta, Surakarta, Central Java, Indonesia
[6] Department of Neurology, Faculty of Medicine, Sebelas Maret University, Surakarta, Central Java, Indonesia
[7] Department of Anatomy and Embryology, Faculty of Medicine, Sebelas Maret University, Surakarta, Central Java, Indonesia

Corresponding author
Flora Ramona Sigit Prakoeswa,
frsp291@ums.ac.id

## ABSTRACT

**Introduction:** Involvement of a chemokine known as C-X-C motif chemokine ligand 10 or CXCL10 in the immunopathology of leprosy has emerged as a possible immunological marker for leprosy diagnosis and needed to be investigate further. The purpose of this systematic review is to assess CXCL10's potential utility as a leprosy diagnostic tool and evaluation of therapy.

**Methods:** This systematic review is based on Preferred Reporting Items for Systematic Reviews and Meta-Analyses (PRISMA) 2020. A thorough search was carried out to find relevant studies only in English and limited in humans published up until September 2023 using PubMed, Scopus, Science Direct, and Wiley Online Library database with keywords based on medical subject headings (MeSH) and no exclusion criteria. The Newcastle-Ottawa Scale (NOS) was utilized for quality assessment, while the Risk of Bias Assessment tool for Non-randomized Studies (RoBANS) was utilized for assessing the risk of bias. Additionally, a narrative synthesis was conducted to provide a comprehensive review of the results.

**Results:** We collected a total of 115 studies using defined keywords and 82 studies were eliminated after titles and abstracts were screened. We assessed the eligibility of the remaining 26 reports in full text and excluded four studies due to inappropriate study design and two studies with incomplete outcome data. There were twenty included studies in total with total of 2.525 samples. The included studies received NOS quality evaluation scores ranging from 6 to 8. The majority of items in the risk bias assessment, using RoBANS, across all included studies yielded low scores. However, certain items related to the selection of participants and confounding variables showed variations. Most of studies indicate that CXCL10 may be a helpful immunological marker for leprosy diagnosis, particularly in leprosy reactions as stated in seven studies. The results are better when paired with other immunological

markers. Its effectiveness in field-friendly diagnostic tools makes it one of the potential biomarkers used in diagnosing leprosy patients. Additionally, CXCL10 may be utilized to assess the efficacy of multidrug therapy (MDT) in leprosy patients as stated in three studies.

**Conclusion:** The results presented in this systematic review supports the importance of CXCL10 in leprosy diagnosis, particularly in leprosy responses and in tracking the efficacy of MDT therapy. Using CXCL10 in clinical settings might help with leprosy early diagnosis. Yet the findings are heterogenous, thus more investigation is required to determine the roles of CXCL10 in leprosy while taking into account for additional confounding variables.

# INTRODUCTION

Chronic bacterial infection known as Leprosy, sometimes known as Hansen's disease, is caused by a gram-positive obligate and acid-fast bacilli that lives inside cells and attracts phagocytes in peripheral nerves and Schwann cells in the skin called *Mycobacterium leprae* (*M. leprae*) (*Chen et al., 2022*). It possesses a characteristic known as acid-fastness, which enables it to resist color fading induced by acid during staining processes, causing it to appear red when observed under a microscope (*Prakoeswa, Rumondor & Prakoeswa, 2022*). *Mycobacterium leprae* usually targets skin and peripheral nerves leading to neuropathy, persistent abnormality, and impairment (*Alinda et al., 2020*). Prolonged close contact with untreated leprosy patients can cause transmission and most likely transmits through upper respiratory secretions (*Alinda et al., 2020*; *Maymone et al., 2020*). Leprosy was reported in 135 WHO Member States in 2021, accounting for 133.781 confirmed cases and 140.546 new cases (*World Health Organization, 2021*). India has the highest number of new leprosy cases, with total of 134,752 cases, followed by Brazil with 33,303 cases. Indonesia comes in third place with 16,825 cases and a disability rate of 6.82 people per million population (*Prakoeswa et al., 2022*; *Fatmala, 2016*; *Prakoeswa et al., 2020*; *Endaryanto, Prakoeswa & Prakoeswa, 2020*). These disabilities happen as a result of leprosy patients being reluctant to seek treatment or seeking it later than necessary (*Muhlisin, 2017*). Leprosy manifests in a variety of ways, from localized to generalized disease depending on the host immunological response which suggests immunological involvement in the course of the disease (*Nath, 2016*). Due to immune system dysregulation, which made *M. leprae* more contagious, those who lived in endemic areas were more susceptible to leprosy (*Prakoeswa et al., 2021*).

Innate and adaptive cytokines like IL-12 and IFN-γ, as well as other pro-inflammatory cytokines like TNF-α, are secreted by CD4+ Th1 cells during protective immune responses against *M. leprae* (*Geluk, 2013*). Th1, Th2, Treg, and Th17 cells are four T cell subsets that have been found to be involved in adaptive immune response and increasing the susceptibility of the host to leprosy (*Prakoeswa et al., 2020*; *Prakoeswa et al., 2022*). Both

TNF-α and IFN-γ have been found to attach to the macrophage cellular receptors in leprosy, resulting in creation of NO and free radicals that kill *M. leprae* (*de Sousa, Sotto & Quaresma, 2017*). When pro-inflammatory conditions exist, a variety of cell types including leukocytes, monocytes, fibroblasts, activated neutrophils, epithelial cells, endothelial cells, mesenchymal cells, stromal cells, and keratinocytes release C-X-C motif chemokine ligand 10 or CXCL10, also known as interferon gamma induced protein (IP-10), in response to IFN-γ (*Makarem et al., 2024*; *Ferreira et al., 2021*). However, an increase in CXCL10 is not always followed by an increase in IFN-γ. This may occur because interferon transcription happens first and then declines rapidly, while CXCL10 expression is slower and remains elevated for some time. Some individuals have also been found to be unable to produce CXCL10 with normal interferon levels in mycobacterium infection (*Scollard et al., 2011*).

Along with NF-κB, which controls the expression of genes critical for both innate and adaptive immune responses, TNF-α may also play a role in the induction of CXCL10 (*Hadi et al., 2021*; *Jin et al., 2017*). It causes tissue injury by attracting macrophages, dendritic cells, natural killer (NK) cells, and activated T lymphocyte cells to inflammatory areas (*Ferreira et al., 2021*; *Pujiastuti, Agusni & Rahmadewi, 2017*). CXCL10 can recruit effector Th1 cells to the site of delayed type hypersensitivity and previously has been identified in tuberculoid lesions of leprosy patients (*Stefani et al., 2009*). Consequently, it is possible that CXCL10 attracts Th1 cells to the site of inflammatory reactions in the skin (*Stefani et al., 2009*; *Luo et al., 2021*). Moreover, *M. leprae* has been discovered to trigger innate receptors, including TLR4, *via* PGL-1 and leading to the abnormal production of interferon β (IFN-β), CXCL10, and inducible nitric oxide synthase (iNOS) (*Berto da Silva Prata et al., 2020*). When activated, CXCL10 modulated the formation and development of different types of granulomas and spectrum disease during the evolution of leprosy (*Medeiros et al., 2015*; *de Souza et al., 2016*). CXCL10 is typically elevated in response to *Mycobacterium leprae* infection, so it has potential to be a biomarker for leprosy (*Geluk, 2018*). Although, patients with psoriasis, rheumatoid arthritis with interstitial lung disease, viral meningitis, and other mycobacterial diseases like tuberculosis and sarcoidosis also have been discovered to express CXCL10. This suggest that CXCL10 is elevated in chronic inflammatory disease (*Makarem et al., 2024*; *Ferreira et al., 2021*; *Hungria et al., 2017*).

Studies discovered that CXCL10 was expressed in borderline tuberculoid (BT), mid-borderline (BB), borderline lepromatous (BL), and lepromatous (LL) leprosy skin lesions with varying levels and also increased in paucibacillary (PB) leprosy patients's plasma (*Ferreira et al., 2021*; *Sharma et al., 2015*). In PB leprosy patients and healthy household contacts who are frequently exposed to *M. leprae*, IFN-γ production is stimulated which results in release of CXCL10 (*Freitas et al., 2015*). With varying levels of CXCL10, it's potentially useful to help differentiate types of leprosy (*Sharma et al., 2015*). Besides that, CXCL10 has been found as a possible marker for tracking the effectiveness of treatment in people with multibacillary leprosy after the introduction of multidrug treatment (MDT) (*Ferreira et al., 2021*). More severe types of the disease are frequently linked to higher levels of CXCL10, which has been discovered to be helpful in identifying T1R reactions in cases of borderline leprosy, as T1R reaction have been found to have

higher levels of CXCL10 (*Freitas et al., 2015*; *Geluk, 2018*). According to research, CXCL10 is an immune marker also linked to neural leprosy despite not being explicitly linked to fibrosis in neural leprosy (*Medeiros et al., 2015*).

Preliminary studies has been showed that CXCL10 is associated with a variety of human diseases including infectious diseases, chronic inflammatory disease, and immune dysfunctions (*Mertaniasih et al., 2021*). This makes CXCL10 currently considered as a diagnostic tool for leprosy, because of its role in their pathogenesis and easily accessible detection method (*Pujiastuti, Agusni & Rahmadewi, 2017*; *Medeiros et al., 2015*; *Mertaniasih et al., 2021*). Identifying CXCL10 can be convenient through easily accessible lateral flow assays, rendering it a feasible indicator for detecting leprosy (*Van Hooij et al., 2016*). Presence of CXCL10 in leprosy patients can help establish a diagnosis apart from using clinical diagnosis criteria (*Sharma et al., 2015*; *Rawat & Chahar, 2016*). Therefore, this systematic review was conducted in order to draw conclusions from all studies about the relevance of CXCL10 in the diagnosis of leprosy, determining the types of leprosy, and as indicator of the efficacy of leprosy therapy.

# SURVEY METHODOLOGY

## Study design

This systematic review is based on Preferred Reporting Items for Systematic Reviews and Meta-Analyses (PRISMA) 2020 and the protocol was registered in PROSPERO (ID: CRD42023460378). Five international databases including PubMed, Scopus, Science Direct, and Wiley Online Library were used to conduct the search in October 2023. Keywords based on medical subject headings (MeSH) were used in the search of relevant scientific studies. The keywords used are: (CXCL10 OR "CXC Chemokine Ligand 10" OR IP#10 OR "Interferon#Inducible Protein 10" OR "Small Inducible Cytokine B10" OR "IFN#gamma#Inducible Protein#10" OR gammaIP#10 OR "Chemokine (C-X-C Motif) Ligand 10" OR "Chemokine (CXC Motif) Ligand 10" OR "Interferongamma#Inducible Protein of 10") AND (Leprosy OR "Hansen disease" OR "Hansen's disease" OR "Morbus Hansen" OR Leprae) AND (Immunological Marker OR Marker$ OR Biomarker$ OR Chemokine OR Immunology OR "Immunologic Marker" OR "Immunological Marker$"). We did not place any limitations on publishing type or date.

## Inclusion and exclusion criteria

Using rayyan.ai (https://www.rayyan.ai/), we performed an initial title and abstract screening. We selected studies limited in humans with data on CXCL10 presence in leprosy patients' serum up to 30[th] of September 2023. We only selected papers in English. This systematic review didn't have any exclusion criteria for the studies used.

## Study selection

Results from the initial search of studies were uploaded to rayyan.ai, which is a free website that could be accessed freely. Rayyan.ai helped organizing these manuscripts based on their tags. Then, we search if there are any duplicates. After the duplicates were deleted, we separate these manuscripts into included and not included based on its title, abstract, and

full text. Two reviewers, F. R. S. P and N. H., evaluated the full texts, abstracts, and titles of the studies based on the inclusion and exclusion criteria. A third party (L. M. D.) would settle any differences amongst all reviewers.

## Quality assessment

We utilized the Newcastle-Ottawa Scale (NOS) to assess the quality of the selected studies. The NOS evaluates the studies based on selection, comparability, and outcome, using seven questions for cross-sectional studies and eight questions for case-control and cohort studies with maximum score of nine points. First, we included all cross-sectional, case-control, and cohort studies we find as NOS is used for these types of studies. Then, we read the manuscript thoroughly to judge and score these studies based on questions provided from the scale. Studies were categorized as high-quality if they scored six or more points. The critical judgments were made by two reviewers (F.R.S.P. and N.H.), and in cases of disagreements throughout the process, a third reviewer (L. M. D.) was consulted.

## Risk bias assessment

In this study, the Risk of Bias Assessment tool for Non-randomized Studies (RoBANS) questionnaire, consisting of six items, was employed to assess the risk of bias in the selected studies. This questionnaire evaluated various aspects, including participant's selection, consideration of confounding variables, exposure measurement, blinding of outcome evaluations, handling of incomplete outcome data, and selective outcome reporting, each rated as low, unclear, or high risk. The study outcomes were visually presented using graphs, including a traffic-light plot and a summary plot. Because there are many aspects that influence the results, it will make the results not objective and unreliable if you do not consider these aspects. With unreliable results, this research couldn't be used as reference. Similar to the previous process, two reviewers (F.R.S.P. and N.H.) conducted this evaluation, and in cases of disagreements, a third reviewer (L. M. D.) was involved.

## Data extraction and analysis

The data analysis used in this study involved predefined sheets that included information on study design, date and geographic location of testing, aim of study, population description and setting, inclusion and exclusion criteria, diagnostic methods and findings, tools for measurements, classification of leprosy, treatments, and outcomes. This analysis was conducted by three reviewers (N. H., E. J. H., and M. F. I), and a qualitative method was employed to assess the collected data.

# RESULTS

## Study selection

The study selection process of this study followed the PRISMA guidelines. Four databases (PubMed = 33; Scopus = 54; Wiley Online Library = 21; Science Direct = 7) collected a total of 115 studies retrieved from these databases. Reviewers removed seven duplicated records before the screening process. Based on the screening of titles and abstracts, we excluded 82 records. We assessed the eligibility of the remaining 26 reports in full text and
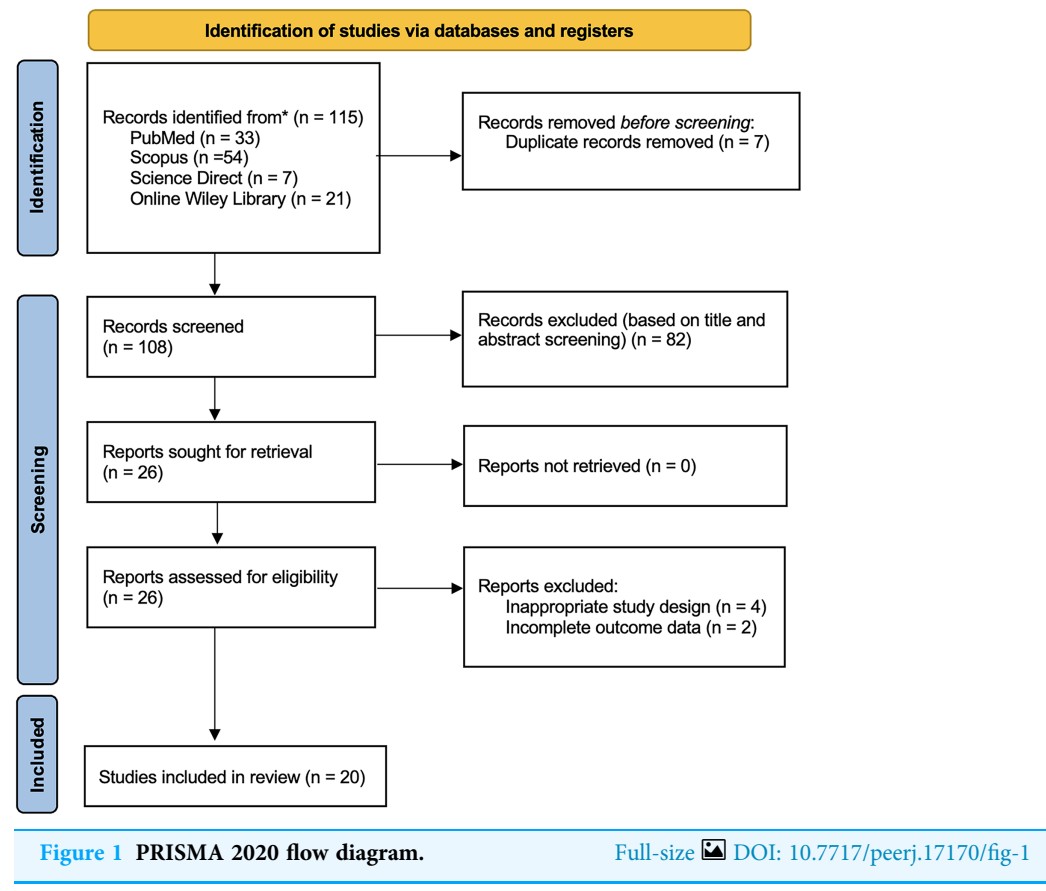

**Figure 1  PRISMA 2020 flow diagram.**     

excluded four studies due to inappropriate study design and two studies with incomplete outcome data. The final analysis included twenty articles in the qualitative synthesis of this review shown in Fig. 1.

## Quality and risk of bias

We utilized The Newcastle-Ottawa Scale (NOS) and The RoBANS tool in the assessment process to evaluate the quality and risk of bias in the selected studies. The results of the quality assessment can be found in Table 1. The included studies scored between six and eight points in the quality assessment, and as a result, they were considered high-quality studies. The results of the risk of bias assessment using the RoBANS tool are depicted in Figs. 2 and 3. While the majority of the items received "low" scores, certain items related to selection of participants, confounding variables, measurements of exposures, and incomplete results data were rated as "unclear".

## Study characteristics

There are 20 articles included in this study. Out of twenty articles, fifteen were cross-sectional studies, two were case control studies, and three were cohort studies. Samples from these included articles ranged from 8 to 715, with total of 2.525 samples. Research was conducted across multiple nations. Table 1 provides a detailed overview of the study's characteristics.

**Table 1 Study characteristics.**

| No | Author (year) | Country | Study design | CXCL10 measurement | Time of CXCL10 measurement |
|---|---|---|---|---|---|
| 1 | *Chaitanya et al. (2013)* | India | Cross-sectional | Incisional skin biopsies for RNA extraction and PCR studies, also ELISA testing of peripheral venous blood and urine samples. | At admission |
| 2 | *van Hooij et al. (2018)* | Brazil, China, and Ethiopia | Cross-sectional | PGL-1 IgM, IP-10, CCL4, and CRP levels with UCP Conjugates and LF Strips | At the time of the first diagnosis, before the start of multidrug therapy (MDT). |
| 3 | *Bobosha et al. (2014)* | Ethiopia and Netherlands | Cross-sectional | Blood tested for combined diagnostic value of IP-10, IL-10, and anti-PGL-I antibodies using ELISA and UCP-LFA | At admission |
| 4 | *Scollard et al. (2011)* | India and United States of America | Cohort | Blood and skin biopsies were drawn and CXCL10 levels were measured with ELISA, real-time PCR, and immunostaining | Blood : Before, during, and after evidence of T1R. Biopsy : at diagnosis and follow-up visits. |
| 5 | *Sharma et al. (2015)* | India | Cross-sectional | Skin biopsies and RNA was measured for CXCL10 with ELISA. Biopsies evaluated with IHC staining. | At admission |
| 6 | *Van Hooij et al. (2016)* | Bangladesh | Cross-sectional | Bloods was drawn and detected for IL-10, IP-10, CCL4 and PGL-I with ELISA and UCP-LFA. | At admission |
| 7 | *Corstjens et al. (2019)* | Netherlands, Brazil, and South Africa | Cross-sectional | Blood was drawn and PGL-I IgM antibodies, IP-10, and CRP measured with UCP-LFA. | At admission |
| 8 | *Khadge et al. (2015)* | Bangladesh, Brazil, Ethiopia, and Nepal | Cohort | Blood was drawn and analized serogically with ELISA. | Newly diagnosed and untreated patient without reaction (t = 0), before anti-reactional therapy and new RR patient (t = x), following MDT and/or steroid therapy (t = end), and after both. |
| 9 | *Ferreira et al. (2021)* | Brazik | Cross-sectional | Assess gene expression and cytokines in serum and whole blood using ELISA and RT-PCR. | At time of diagnosis, before treatment, and after treatment was finished (released). |
| 10 | *Hungria et al. (2017)* | Brazil | Cross-sectional | Whole blood measured for CXCL10 and IFN-γ serum levels using whole blood assay (WBA) and ELISA. | At admission |
| 11 | *Queiroz et al. (2021)* | Brazil | Cross-sectional | Chemokines CXCL8 (IL-8), CCL2, CXCL9, and CXCL10, as well were tested in all patients after blood was obtained. | First visit in 2014 (Time 0-T0) and second visit in 2015 (Time 1-T1). |
| 12 | *Angst et al. (2020)* | Brazil | Cross-sectional | Sensory nerve was biopsied, then serum cytokine levels and histopathological evaluation were done. A clinical and neurophysiological evaluation was also conducted. | At admission |
| 13 | *Cunha et al. (2023)* | Brazil | Cross-sectional | Whole blood utilized for TLR4 genotyping by PCR and chemokine and cytokine measurement using cytometric beads array. | At admission |
| 14 | *van Hooij et al. (2019)* | Bangladesh | Cross-sectional | Blood measured for leprosy biomarkers using multiples bead arrays (MBA), ELISA, and lateral flow assays (LFA) strips. | At admission |
| 15 | *Das et al. (2023)* | India | Cross-sectional | Skin biopsies from skin lesions utilized for multiplex qPCR, RT2 PCR profiler arrays, and whole transcriptome hybridization arrays. | At admission |

(Continued)

| No | Author (year) | Country | Study design | CXCL10 measurement | Time of CXCL10 measurement |
|---|---|---|---|---|---|
| | | | | **Table 1 (continued)** | |
| 16 | *van Hooij et al. (2021)* | Bangladesh and South Korea | Cross-sectional | Fingerstick blood test sample were collected and tested for levels of biomarker using multi-biomarker test (MBT). Also, whole blood assay was conducted. | At admission |
| 17 | *Medeiros et al. (2015)* | Brazil | Cross-sectional | Mouse monoclonal antibodies against CCL2 used to stain CCL2 and counterstained with Mayer's hematoxylin and mounted. Immunoreactivities was observed under the microscope. | At admission, 6 months MDT for PB, and 12 months MDT for MB. |
| 18 | *Geluk et al. (2010)* | Netherlands | Cross-sectional | Whole blood incubated in 48-well plate with antigen, then flow cytometry was performed. | At admission |
| 19 | *de Santana et al. (2017)* | Brazil | Case control | Whole blood was centrifuged extract serum. ELISA chemokine tests were then used to quantify MCP-1 levels. | At admission |
| 20 | *Stefani et al. (2009)* | Brazil | Crase control | Blood was drawn into EDTA and centrifuged right away. Samples of EDTA plasma were thawed and centrifuged, and the supernatant was filtered. | At admission |

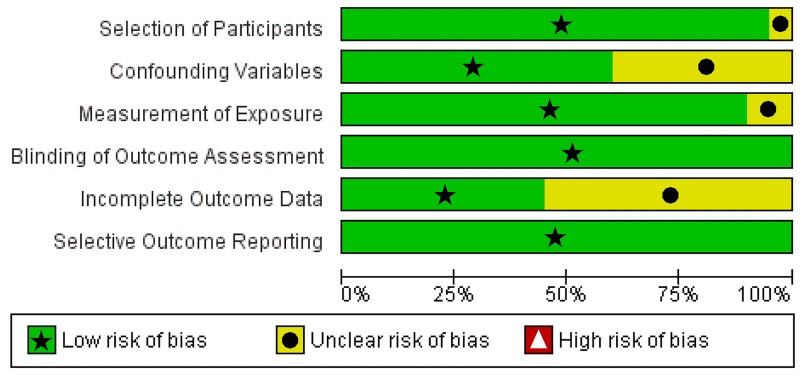

**Figure 2 Risk of bias assessment tool for non-randomized studies (RoBANS) graph.**

## Study results

### CXCL10 in leprosy reaction

A cross-sectional study by *Chaitanya et al. (2013)* in 100 leprosy patients with Type 1 reaction (T1R) and without reaction (NR) investigated the mRNA expression ratios of CXCL10 serum levels. This study found significantly higher levels of CXCL10 ($p < 0.05$) in skin lesion of leprosy patients with Type 1 reaction (T1R) compared to no reaction cases (NR). Findings by *Chaitanya et al. (2013)* was supported by a cohort study conducted by *Scollard et al. (2011)* which evaluated the correlation of CXCL10 with Type 1 reaction (T1R). This study examined samples across time with monthly follow-up and discovered

| | Selection of Participants | Confounding Variables | Measurement of Exposure | Blinding of Outcome Assessment | Incomplete Outcome Data | Selective Outcome Reporting |
|---|---|---|---|---|---|---|
| Bobosha et al (2014) | ? | ? | + | + | + | + |
| Chaitanya et al (2013) | + | + | ? | + | + | + |
| Corstjens et al (2019) | + | ? | + | + | + | + |
| Cunha et al (2023) | + | ? | + | + | ? | + |
| Das et al (2023) | + | ? | + | + | ? | + |
| Ferreira et al (2021) | + | + | + | + | + | + |
| Geluk et al (2010) | + | + | + | + | ? | + |
| Hungria et al (2017) | + | + | + | + | + | + |
| Khadge et al (2015) | + | ? | + | + | ? | + |
| Medeiros et al (2015) | + | ? | + | + | + | + |
| Moraes Angst et al (2020) | + | + | + | + | ? | + |
| Queiroz et al (2021) | + | + | + | + | ? | + |
| Santana et al (2017) | + | + | + | + | ? | + |
| Scollard et al (2011) | + | + | + | + | ? | + |
| Sharma et al (2015) | + | ? | + | + | ? | + |
| Stefani et al (2009) | + | + | + | + | ? | + |
| van Hooij et al (2016) | + | + | ? | + | + | + |
| van Hooij et al (2018) | + | ? | + | + | + | + |
| van Hooij et al (2019) | + | + | + | + | + | + |
| van Hooij et al (2021) | + | + | + | + | ? | + |

**Figure 3 Summary of risk of bias assessment tool for non-randomized studies (RoBANS).**

that in patients with BL ($p < 0.0006$) and BT ($p < 0.0001$), CXCL10 was considerably higher during T1R. But, CXCL10 and the severity of T1R did not have any correlation, as indicated by the value of $p = 0.85$ ($p > 0.05$). Individuals with T1R had greater CXCL10 mRNA median levels in their biopsy specimens than do individuals without T1R ($p < 0.02$). It was also stated that there was no discernible relationship between

intralesional levels of CXCL10 and IFN-γ mRNA (*Scollard et al., 2011*). But, a cross-sectional study by *Geluk et al. (2010)* discovered that CXCL10 is secreted in response of IFN-γ, suggesting a correlation between the levels of CXCL10 and IFN-γ. *Das et al. (2023)* in a cross-sectional study also found significantly different CXCL10 expression between Type 1 reaction group (T1R) and without reaction group (NR) with $p$ value of 0.005, where its expression was higher in T1R. Meanwhile, CXCL10 expression wasn't statistically different between NR *vs* T2R or T1R *vs* T2R (*Das et al., 2023*).

Significant result on understanding CXCL10's potential as a biomarker for distinguishing between reactional and non-reactional leprosy was also found in cross-sectional study by *Sharma et al. (2015)*. In this study, CXCL10 mRNA levels were consistently high across all leprosy subgroups, but reaching >500 times in BB-T1R. Differences in CXCL10 levels was seen between borderline leprosy (BB) and BB-T1R ($p < 0.05$). Compared to non-reactional BB leprosy, T1R biopsies showed more pronounced CXCL10 staining. Only the difference between BB and BB-T1R was statistically significant ($p < 0.01$), meanwhile differences between BT/BT-T1R and BL/BL-T1R wasn't significant (*Sharma et al., 2015*). *Khadge et al. (2015)* findings in a cohort study aiming to identify specific biomarker of inflammatory reactional episodes of leprosy in four different countries (Bangladesh, Brazil, Ethiopia, and Nepal) supported previous studies. The levels of CXCL10 significantly peaked at the start of a Type 1 reaction ($p = 0.0059$). Patients with T1R also had greater CXCL10/IL-10 ratios than those without T1R, although the difference was not statistically significant (AUC: 0.79; range: 0.955-1). There was no discernible difference in the levels of CXCL10 between patients with no reactions and healthy donors serving as controls, suggesting CXCL10 may serve as a marker for T1R. *Stefani et al. (2009)* also found elevation of CXCL10 level in plasma of T1R patients. *Ferreira et al. (2021)* supported this by stating that CXCL10 plasma concentration differences in T1R patients and control was found highly significant with $p = 0.004$.

### CXCL10 as diagnostic properties

Study by *Queiroz et al. (2021)* discovered a decrease in CXCL10 expression in paucibacillary (PB) patients following therapy. This cohort also found that CXCL10 levels were greater in the multibacillary (MB) group than the household contact (HHC) during the first visit. But unlike *Queiroz et al. (2021)*, cross-sectional study by *Cunha et al. (2023)* found household contact (HHC) of paucibacillary (PB) patients have higher level of CXCL10 compared to HHC of multibacillary (MB) patients with $p < 0.05$ in *M. leprae*-stimulated culture. This study showed higher levels of chemokine, like CXCL10, along with increasing IL-10 in HHC of PB patients, based on chemokine signature of *M. leprae*-stimulated culture. For the AG genotype, higher amounts of CXCL10 were discovered in the unstimulated cultures. Increased level of CXCL10 was associated with A allele and G allele. TLR4 rs1927914 genotypes showed an association between AG (one of them CXCL10) and AA (CXCL8, CCL2, TNF, and IL-2) genotypes with a more pronounced secretion of chemokines and cytokines. *de Santana et al. (2017)* also found out about the difference in production of CXCL10 in polymorphisms of genes TLR1, 2 and 4. Carriers of

the G allele in TLR2_ rs7656411, such as genotypes GG, GT and TT, produced higher levels of CXCL10 (*de Santana et al., 2017*).

A field-friendly diagnostic assay was used in a cross-sectional investigation by *Bobosha et al. (2014)* to assess combined cytokine profiles in responses to *M. leprae* antigen. When exposed to *M. leprae* antigen, CXCL10 levels were observed to be greater, and patient CXCL10/IL-10 ratios were noticeably higher than those in EC. The level of CXCL10 was substantially higher ($p = 0.02$) in ML2478-stimulated samples compared to EC, and it worked more quickly than IFN-γ within the same time frame. CXCL10 results from both wet and dry assays had strong correlation ($R^2$ 0,790), indicating that the dry-format CXCL10-UCP-LFA was appropriate for field use. *Van Hooij et al. (2016)* also identified IP-10 profiles in *M. leprae* infection and type of leprosy in Bangladesh using UCP-LFA. This cross-sectional study found the correlations of UCP-LFA and ELISA were significant for IL-10, CXCL10, CCL4, and anti-PGL-I-antibodies ($p < 0.0001$), proving UCP-LFA's performance was in par with ELISA. Based on AUC, CXCL10 could differentiate MB patients from endemic control or EC ($p \leq 0.05$, $p \leq 0.01$, $p \leq 0.0001$). Concentration of CXCL10 was found higher in HHC with BCG vaccination than HHC without BCG vaccination ($p = 0.018$).

*Hungria et al. (2017)* investigated the use of whole blood assay as diagnostic tool of PB leprosy with detecting CXCL10 serum levels in 156 sample. Higher CXCL10 levels was found in whole blood assay using PB sample compare to sample from EC ($P = 0.001$) and also fair AUC of 0.75 (fair = 0.7–0.8). According to this study, CXCL10 could only distinguish between PB and EC in samples containing recombinant protein (ML0276 + LID-1 WBA), and it was unable to distinguish between active PB and *M. leprae*-infected people (HHC). CXCL10 alone did not distinguish PB from EC as well as IFN-γ. Additionally, neither CXCL10 nor IFN-γ could distinguish PB from HHC. To contrast PB and EC, however, CXCL10 and IFN-γ have a higher sensitivity (89%), but a lower specificity (compared to using each molecule alone).

A cross-sectional study by *van Hooij et al. (2018)* described the evaluation of multiple UCP-LFAs in measuring CXCL10 as one of biomarker for diagnosing leprosy in leprosy patients, household contact (HHC), and endemic control (EC). According to this study, in both low- and high-endemic communities, CXCL10 was the most important cellular marker for identifying leprosy patients with LL/BL and BT/TT. In Brazil (hyperendemic), CXCL10 levels between LL/BL patient *vs* HHC and EC patients significantly different demonstrated by notable AUC values. In China (low endemic), CXCL10 levels were significantly different in LL/BL patients compared to HHC and EC. The levels of CXCL10 were significantly different in BT/TT compared to HHC and EC. Also, CXCL10 levels could be used to separated EC from BT/TT patients in Brazil and China. But, the use of other biomarkers like PGL-I antibodies and CCL4 increased the sensitivity of the test.

Result from cross-sectional study by *Corstjens et al. (2019)* supported *van Hooij et al. (2019)* where CXCL10 proved to be detected using FSB sample with significant correlation between serum and FSB-values ($p < 0.0001$, reasonable $R^2 = 0.61$) and showed significantly higher levels of IP-10 in leprosy patient compared to controls. But, *in vivo* concentration of CXCL10 is relevantly lower than generally found for CRP so CXCL10 FSB assay performed

with higher sample load. Fresh FSB is the chosen sample for CXCL10 because frozen CXCL10 FSB samples could not be tested. *van Hooij et al. (2019)* discovered that CXCL10 levels were a marker of MB leprosy and that they were only substantially different from EC in the MB group. Patients with MB and PB both have higher levels of CXCL10. Similar outcomes were obtained using fingerstick blood for UCP-LFA, and CXCL10 continued to be an important marker in plasma for MB patients, which means we could use fingerstick blood as a sample in diagnosis of leprosy.

*van Hooij et al. (2021)* also found that in Bangladesh, CXCL10 detection using multi-biomarker test (MBT) needed 10-fold dilutions, unlike CRP and PGL-I antibodies that could detected using 1,000-fold dilutions. With single strip UCP-FLA, NUM score showed that it could accurately differentiate leprosy patients from endemic control with AUC 0.93. The NUM score shown a similar number in the MBT results (AUC: 0.9: $p < 0.0001$), indicating the ability of MBT to identify leprosy patients in endemic locations. In South Korea, it was discovered that MBT could tell leprosy patients apart from nearby healthy controls (AUC: 0.88; $p < 0.0001$). This demonstrates how MBT, of which CXCL10 was a component, could provide a qualitative result (positive or negative) for each biomarker and enable in-depth biomarker assessment. Another good option for a more practical technique to get blood for MBT was to use fingerstick blood.

### CXCL10 and neuropathy

From histopathological findings in study by *Medeiros et al. (2015)*, Schwann cell immunoreactivity toward CXCL10 was found. Along the longitudinal portion of the PNL AFB+ nerve, inflammatory infiltrating cells that resembled the morphology of PNL AFB- nerve macrophages and lymphocytes. In 12 of the PNL nerves, CXCL10 was detected (66.7% in PNL AFB+ & 72.7% in PNL AFB-). Out of the six AFB- PNL nerves with epithelioid granuloma, two nerves (66.6%) had a moderate amount of Schwann and inflammatory cells immunolabeled with CXCL10. In two of the six AFB- PNL samples with a mononuclear cell infiltrate, Schwann cells and macrophages displayed moderate to strong CXCL10-immunoreactivity. In AFB- PNL samples lacking a leukocytic infiltrate, Schwann cells and endothelial cells showed moderate to weak CXCL10-positivity during histopathologic analysis. There was no CXCL10 and CCL2 expression found in the nerve sections of non-leprosy group. A cross-sectional study by *Angst et al. (2020)* found that diabetes group with diabetic neuropathic pain have higher concentration of CXCL10 compared to painless leprosy neuropathy and leprosy with neuropathy. Meanwhile, painless leprosy neuropathy had significantly higher concentration of CXCL10 than leprosy patients with neuropathy ($p = 0.02$). In patients with neuropathic pain, CXCL10 concentration were slightly elevated compared to those with neuritis (*Angst et al., 2020*).

### CXCL10 and leprosy's therapy

Study by *Queiroz et al. (2021)* showed decreasing CXCL10 production in PB, MB, and HHC following treatment. Decreasing expression of CXCL10 after MDT is supported by a cross-sectional study by *Ferreira et al. (2021)*. It was found after 12 doses of MDT, production of CXCL10 in skin lesion based on gene expression ($p = 0.008$) and

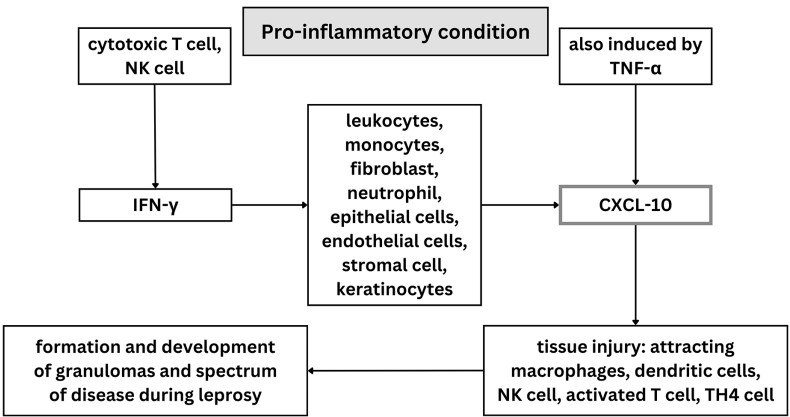

**Figure 4  Illustration of role of CXCL-10 on leprosy.**

histopathology was significantly reduced. Following MDT, serum levels of CXCL10 dropped ($p = 0.02$, $p \leq 0.01$). But, bacilloscopic index reduction did not coincide with a significant drop in CXCL10. The group with lower bacilloscopic index after 12 doses of MDT (WR) demonstrated an increase in CXCL10 levels in cells triggered by *M. leprae* in comparison to cells without stimulation ($p \leq 0.05$). This finding was also supported by *Khadge et al. (2015)* where with the start of anti-reactional treatment, serum levels of CXCL10 dropped ($p = 0.002$) which suggest the correlation of CXCL10 levels with T1R.

## DISCUSSION

Twenty articles were included in the qualitative synthesis process of this systematic review, which evaluated the potential contribution of CXCL10 in diagnosing leprosy and leprosy therapy. The scope of this study investigation into CXCL10's function in leprosy diagnosis was restricted to humans. Included studies used in this systematic review were investigating CXCL10 levels in different types of leprosy, between leprosy patients and control, between leprosy with reaction and without reaction, before and after MDT treatment, and in field friendly diagnostic tools. Some studies have measured CXCL10 levels using ELISA, PCR, and UCP-LFA; other studies used IHC staining on biopsy samples. In general, illustration of role of CXCL-10 on leprosy was visualized on Fig. 4.

### CXCL10 as diagnostic properties

Constant spread of *M. leprae* has led leprosy researchers to concentrate more on finding biomarkers to aid in diagnosis. As a result, there is an increasing demand for novel, sensitive diagnostic instruments based on particular biomarkers that, ideally, translate easily into tests that are suitable for the field (*Van Hooij et al., 2016*). Expression of CXCL10 were found to be higher in leprosy patients than control or household contact (*Ferreira et al., 2021*; *Queiroz et al., 2021*; *Cunha et al., 2023*). *Queiroz et al. (2021)* stated greater levels of CXCL10 levels in multibacillary (MB) group than the household contact (HHC). It was found that in comparison to paucibacillary contacts, *M. leprae*-induced IFN-γ production is greater in multibacillary contacts. People living in endemic environments are more vulnerable to leprosy because they are in close proximity to others

who have the disease, particularly those who have the multibacillary (MB) form (*Prakoeswa et al., 2021*). Studies have shown that leprosy tends to affect those living in close proximity more often than those living apart, corroborating the higher risk of contracting the disease for individuals who have spent more than four years near leprosy patients (*Prakoeswa et al., 2020*, *2021*). Additionally, *M. leprae*-induced IFN-γ production is stronger in multibacillary contacts than in paucibacillary contacts (*Cunha et al., 2023*). The more severe the leprosy, the higher the CXCL10 levels (*Freitas et al., 2015*). CXCL10 levels increased in all types of leprosy, but they were higher in patients with a reaction where usually happened in multibacillary leprosy (*Sharma et al., 2015*).

*Cunha et al. (2023)* found high levels of CXCXL10 in household contact (HHC) of PB and MB, but higher HHC of PB patients than HHC of MB. Paucibacillary (PB) patients show antigen-specific T cell responses (Th1 and Th17), where Th1 secrete IFN-γ (*Geluk, 2013*; *Ferreira et al., 2021*). High level of CXCL10 was as response to IFN-γ, where leukocytes, monocytes, activated neutrophils, epithelial cells, endothelial cells, stromal cells, and keratinocytes, release CXCL10 (*Geluk, 2013*; *Corstjens et al., 2019*; *Bobosha et al., 2014*). According to histological results, pure neural leprosy (PNL) nerves showed CXCL10 expression. These nerves showed moderate to strong CXCL10-immunoreactivity, which may indicate that this chemokine is involved in initiating the intraneural response. But, CXCL10 were not linked to neural fibrosis (*Medeiros et al., 2015*). Higher levels of CXCL10 were also found to be higher in PB patients than EC and could differentiate PB from EC in samples containing recombinant protein (ML0276 + LID-1 WBA). But, CXCL10 alone did not distinguish PB from EC as well as IFN-γ (*Hungria et al., 2017*). Study by *van Hooij et al. (2018)* in low and high endemic population showed that CXCL10 was the most significant marker to identify BT/TT and LL/BL patients. In low and high endemic countries, CXCL10 could be used to differentiate between LL/BL patients and HHC and EC. Meanwhile in BT/TT patients, CXCL10 could be used to differentiate from endemic controls (*van Hooij et al., 2018*).

Combination of UCP-LFA were useful to detect various analytes, such as proteins, polysaccharides, antigens, drugs, nucleic acid, and antibodies. This instrument was useful and didn't require expensive equipment to operate, making it appropriate for use in leprosy-endemic areas where access to advanced laboratory instruments is frequently limited (*Bobosha et al., 2014*). Multi-biomarker test (MBT) using UCP-LFA strip could accurately differentiate leprosy patients from endemic control in endemic and non-endemic settings (*van Hooij et al., 2021*). Similar test findings were achieved with the six-marker MBT strip as they were with the individual UCP-LFA strips for every biomarker individually in the past, which suggest this tool feasibility as a diagnostic tool (*van Hooij et al., 2019*). In both leprosy-endemic and non-endemic settings, MBT can be helpful as an additional diagnosis for individuals exhibiting signs that could indicate leprosy. The assay time can be greatly reduced and a sample-to-result can be obtained on the same day with the usage of UCP-LFA (*Bobosha et al., 2014*). In addition, the strips can be kept in patients' file and revisited in another time because of the assay components' chemical stability, which offered easier way to monitor patient's profile. With this potential, leprosy detection can be done easily and at low cost, yet it gives reliable results.

Thus, the scope of early leprosy diagnosis can be broader, enabling earlier initiation of leprosy treatment (*Corstjens et al., 2019*; *Bobosha et al., 2014*).

Using UCP-LFA, CXCL10 was higher in response to *M. leprae* antigen and higher CXCL10/IL-10 ratio in leprosy patient compared to endemic control (EC) (*Bobosha et al., 2014*). It was also found, CXCL10 could differentiate MB patients from EC (*Van Hooij et al., 2016*; *van Hooij et al., 2019*) by UCP-LFA sample. *van Hooij et al. (2019)* validated the use of CXCL10 as a biomarker for MB leprosy, which is consistent with earlier research. However, PB patients and HC could not be differentiated given that the indicators displayed comparable responses for CXCL10, particularly in highly endemic areas (*Van Hooij et al., 2016*; *van Hooij et al., 2018*). Patients with PB and HC also exhibit comparable immune responses and frequently have undetectable *M. leprae* bacilli counts (*Van Hooij et al., 2016*). The use of CXCL10 as a biomarker to diagnose leprosy is more effective and sensitive with the combination of other biomarkers like PGL-1 antibodies, IL-10, CRP, and CCL4 (*van Hooij et al., 2018*, *2021*, *2019*; *Van Hooij et al., 2016*). Combining anti-PGL-I IgM, IL-10, and CXCL10 was suggestive for leprosy classification, allowing MB and PB patients to be distinguished from one another. A plasma biomarker profile that included αPGL-I IgM, CXCL10, S100A12, ApoA1, and CRP accurately diagnosed leprosy patients of any kind with high sensitivity (86%) and specificity (90%) in the UCP-LFAs. This shows that this signature can be used to diagnose leprosy because it can identify patients with both high and low bacillary loads (*Van Hooij et al., 2016*).

To accommodate field-friendly diagnostic tools for leprosy, a study discovered that CXCL10 was detected in fingerstick blood (FSB) sample and showed to be higher in leprosy patients than in control. This suggest the use of FSB sample offered a simpler way of collecting samples and also could be beneficial in diagnosing leprosy (*Corstjens et al., 2019*). This was supported by *van Hooij et al. (2019)* where CXCL10 is an important marker in plasma for MB patients even in FSB samples. Since HHCs of patients with MB leprosy have the highest chance of contracting an *M. leprae* infection, they are good candidates for preventative medication delivery in multiple studies (*Barth-Jaeggi et al., 2016*; *Mieras et al., 2018*; *Ortuno-Gutierrez et al., 2019*; *Richardus et al., 2021*; *Tiwari et al., 2020*). Aimed at slowing the spread of the illness and preventing its progression, the MBT may help identify *M. leprae*-infected people who qualify for post-exposure prophylaxis (PEP), enabling a more effective and focused drug delivery strategy as diagnostic tools in early detection of leprosy (*van Hooij et al., 2021*). To minimize errors in the initial diagnosis of leprosy, one of important steps is the utilization of appropriate tools and parameters. Additionally, confounding variables that may influence the results, such as age and exposure to contacts, should be taken into consideration. It is advisable to do the test not only once but several times to get more reliable results. Furthermore, the results should be evaluated based on the patient's clinical condition.

The detection of CXCL10 in leprosy patient can also be done using rt-PCR, ELISA, whole blood assay, and histopathological evaluation using immunostaining. With rt-PCR, mRNA expression levels of CXCL10 could be detected. It was shown that mRNA expression levels of CXCL10 from skin biopsies of borderline leprosy cases with or without T1R was consistently high. There was significant difference CXCL10 expression between

borderline leprosy (BB) and BB-T1R, where BB-T1R have higher expression than BB (*Sharma et al., 2015*). *Khadge et al. (2015)* used ELISA to identify CXCL10 expression in leprosy patients. Levels of CXCL10 was found to be increased in all types of leprosy, but higher in patient with reaction. This demonstrates the use of rt-PCR and ELISA in diagnosing leprosy reaction (*Sharma et al., 2015*; *Khadge et al., 2015*). *Bobosha et al. (2014)* assess combined cytokine profiles using ELISA in responses to *M. leprae* antigen and found that CXCL10 levels were higher in infected patients than those in EC. *Van Hooij et al. (2016)* also identified CXCL10 profiles in *M. leprae* infection and type of leprosy using ELISA and showed significant levels of CXCL10 in leprosy patients. CXCL10 could differentiate MB patients from endemic control or EC. Using whole blood assay, higher CXCL10 levels was also found in PB sample compared to samples from EC, but couldn't distinguished between PB and HHC (*Hungria et al., 2017*). For immunostaining, biopsy samples showed CXCL10 had more pronounced staining in patient with T1R than in those without reaction (*Khadge et al., 2015*). Compared to non-reactional BB leprosy, T1R biopsies showed more pronounced CXCL10 staining (*Sharma et al., 2015*). In addition, mass spectrometry (MS)-based approaches could be considered to be used in CXCL10 levels detection as it also had the ability to discover disease biomarkers and revealed non-biased profile of protein/metabolites. It has been used in discovering COVID-19 biomarkers (*Zhong, Zhu & Cai, 2021*).

## CXCL10 in leprosy reaction

Type 1 reaction (T1R) in leprosy patients are a delayed type hypersensitivity reaction, caused by exacerbations of immune system (*Chaitanya et al., 2013*). This reaction usually occurred in borderline-borderline leprosy (BB), borderline-tuberculoid leprosy (BT), and borderline-lepromatous leprosy (BL) (*Khadge et al., 2015*). These inflammatory reactions are treated with immunomodulatory medications at high doses, which frequently increase morbidity. They can happen before, during, and after multidrug therapy (MDT) (*Das et al., 2023*). Erythematous and/or oedematous skin lesions with or without ulceration are among the clinical characteristics of T1R, as is oedema of the hands, face, and feet. Even in rare cases, systemic symptoms might result from acute inflammation of the peripheral nerves, which can compromise nerve function (NFI) (*Khadge et al., 2015*; *Chaitanya et al., 2013*). Dermal oedema, oedema, and lymphocytes within the granuloma are the histological characteristics indicative of T1R (*Sharma et al., 2015*).

Patients with T1R have higher mRNA expressions of CXCL10 than patients without reaction (*Chaitanya et al., 2013*). Several studies showed mRNA levels CXCL10 were consistently higher in BL and BT patients during T1R than before T1R, BB patients with T1R compared to those without reaction, as well as in T1R compared to T2R (*Ferreira et al., 2021*; *Scollard et al., 2011*; *Sharma et al., 2015*; *Das et al., 2023*). *Khadge et al. (2015)* supported these findings by conducting a study in four different countries, with levels of CXCL10 peaking at the start of T1R as a result. This means that CXCL10 levels increased in all types of leprosy, but they were higher in patients with a reaction (*Sharma et al., 2015*). Biopsy samples also showed CXCL10 had more pronounced staining in patient with T1R than in those without reaction where immune-expression of CXCL10 was localized in

cytoplasm of macrophages and epithelioid cells (*Sharma et al., 2015*; *Chaitanya et al., 2013*). The higher levels CXCL10 was caused by vital role of proinflammatory cytokines regulations in leprosy reaction, as CXCL10 is expressed in a variety of Th1-type inflammatory disorders due to its' crucial role in drawing activated T cells to tissue inflammation sites (*Scollard et al., 2011*; *Teles et al., 2019*; *Geluk et al., 2014*). Expression of CXCL10 is also induced by IFN-γ, and IFN-γ expression was increased in reactional skin, along with macrophage activation (*Scollard et al., 2011*; *Chaitanya et al., 2013*; *Geluk et al., 2010*). However, there was no correlation of CXCL10 levels with the severity of T1R and CXCL10 couldn't predict the occurence of T1R (*Scollard et al., 2011*).

## Clinical implication

The WHO recommended two standardized multidrug therapy (MDT) regimens for leprosy in 1982. Regardless of the patient's categorization, the uniform therapy for leprosy (U-MDT) is taking clofazimine and dapsone daily for six months, along with monthly rifampicin (*Penna et al., 2017*). A decrease in CXCL10 production was found in PB, MB, and HHC after treatment with MDT. After 12 doses of MDT, production of CXCL10 in skin lesion was reduced, along with serum levels of CXCL10 (*Ferreira et al., 2021*; *Queiroz et al., 2021*). *Khadge et al. (2015)* also discovered similar findings, where serum levels of CXCL10 decreased in the beginning of anti-reactional treatment with corticosteroid. Consequently, the study's findings imply that CXCL10 may be utilized to assess the effectiveness of MDT in patients (*Ferreira et al., 2021*; *Khadge et al., 2015*). It is commonly known that phagocytosis, growth factor and cytokine secretion, and matrix remodeling are just a few of the overlapping processes that make up tissue repair. In this case, tissue repair may be linked to the decrease in proinflammatory mediators in skin cells, such as IFN-γ and CXCL10 (*Ferreira et al., 2021*; *Novak & Koh, 2013*). With the evidence that CXCL10 can assess the effectiveness of MDT, it is also possible to simultaneously monitor drug compliance of leprosy patients. Regular intake of 12 doses of MDT shows a decrease in CXCL10 levels, indicating the potential for CXCL10 to be used in monitoring patient compliance. Further research is needed explore this potential (*Ferreira et al., 2021*; *Queiroz et al., 2021*).

Besides assessing MDT effectiveness' in leprosy patients, CXCL10 could be used as one of diagnostic biomarker for leprosy as described before. There are several studies focusing on using CXCL10 as one of biomarkers use in diagnosing leprosy. These studies also investigated field-friendly diagnostic tool, such as UCP-LFA, to help bigger scale of leprosy diagnosis. All of included studies was conducted in humans. Although there are some studies about CXCL10 as diagnostic biomarkers of leprosy and as MDT effectiveness' parameter, further studies to investigate its' properties in larger populations (endemic and non-endemic settings) with consideration should be conducted. Future research focusing on the development of these tools and CXCL10 as a biomarker in diagnosing different types of leprosy is needed due to the varying outcomes of the included studies, making it difficult to draw certain conclusions.

### Limitation of study

Due to the heterogeneity of data from the included studies and differences in assessed parameters, this systematic review couldn't proceed to a meta-analysis study. Most of the included studies did not address confounding factors in their investigations and did not explain if there was incomplete outcome data. Furthermore, there was no ascertainment of exposure stated in the included case-control and cohort studies. Additionally, some of the included studies did not specify any inclusion or exclusion criteria. Each of the studies included has its own limitations. A relatively modest sample size, primarily resulting from insufficient data or a shortage of blood samples, as well as the loss to follow-up of certain patients, represented one of the study's constraints. The use of non-probabilistic sampling methods, the absence of corrections for confounding variables, and the lack of pre and post-test data were identified as additional limitations in these studies. The involvement of various examiners in data collection also introduced potential bias, contributing to the studies' limitations. To establish the role of CXCL10 as a biomarker for diagnosing leprosy in infected individuals and controls, it is imperative to conduct larger-scale studies that carefully account for confounding variables.

## CONCLUSIONS

In conclusion, the included studies in this systematic review has demonstrated CXCL10's role in diagnosing leprosy, particularly in leprosy reactions. Its effectiveness in field-friendly diagnostic tools makes it one of the potential biomarkers used in diagnosing leprosy patients. Additionally, CXCL10 has the potential to monitor the effectiveness of MDT therapy. However, these findings are heterogeneous, so further research is needed with larger sample sizes and more diverse methods, taking into consideration other confounding factors in order to establish CXCL10 roles in leprosy.

## ACKNOWLEDGEMENTS

The authors acknowledge all the members of Hansen Disease's Research Group (HDRG) of Universitas Muhammadiyah Surakarta for supporting the entire process of this study with conceptualization and peer discussion.

### Funding

The authors received no funding for this work.

### Competing Interests

The authors declare that they have no competing interests.

### Author Contributions

- Flora Ramona Sigit Prakoeswa conceived and designed the experiments, authored or reviewed drafts of the article, and approved the final draft.
- Nabila Haningtyas conceived and designed the experiments, performed the experiments, analyzed the data, prepared figures and/or tables, and approved the final draft.

- Listiana Masyita Dewi conceived and designed the experiments, authored or reviewed drafts of the article, and approved the final draft.
- Ellen Josephine Handoko performed the experiments, analyzed the data, prepared figures and/or tables, and approved the final draft.
- Moch. Tabriz Azenta performed the experiments, authored or reviewed drafts of the article, and approved the final draft.
- Muhana Fawwazy Ilyas performed the experiments, analyzed the data, prepared figures and/or tables, and approved the final draft.

## Data Availability

This is a systematic review.

## Supplemental Information

Supplemental information for this article can be found online at http://dx.doi.org/10.7717/peerj.17170#supplemental-information.

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
