# Peer review of "The role of CXCL10 as a biomarker for immunological response among patients with leprosy: a systematic literature review"

_PeerJ, doi:10.7717/peerj.17170_

## Round 0.1 · original submission · Major Revisions

Please address the queries of both reviewers and amend the manuscript accordingly.

In addition, I note that you used an Artificial Intelligence tool (https://www.rayyan.ai/ in the initial screening steps. Thank you for being transparent about this in line with PeerJ policy (https://peerj.com/about/policies-and-procedures/#publication-ethics) but it is only mentioned in passing in the manuscript so please expand the methodology to fully explain the process.

Reviewer 1 ·

Basic reporting

The authors deal with leprosy, a disease that unfortunately has not yet been eradicated
and that is widespread in various parts of the world. In particular, this review evaluated the potential
contribution of the chemokine CXCL10 in diagnosing leprosy and leprosy
therapy. So the review arouses a lot of interest.
Introduction and background are clear and
also English language is professional. Literature well referenced and relevant.

Experimental design

Research question well defined.

Validity of the findings

The results and discussion are not well clear. Literature data should be represented with Table
reporting patients characteristics, with pie chart showing the % of individuals with positive test for CXCL10
in the respective groups (BT,BB,BL etc.) in the different country, or with graphs reporting the
concentrations of CXCL10 in the respective groups (BT,BB,BL etc.)

Reviewer 2 ·

Basic reporting

no comment

Experimental design

no comment

Validity of the findings

no comment

Additional comments

The role of CXCL10 as a biomarker for immunological response among patients with leprosy: a systematic literature review

Comments on peerj-reviewing-94150-v0
Recommend: Major revision
In this work, the authors mainly reviewed and summarized the literature on CXCL10 in the diagnosis and treatment of leprosy. Through a scientific screening method, the authors selected papers from different countries that are representative and reliable in this respect. The authors explain in detail the possible mechanism of CXCL10 in relation to leprosy in these literatures. In addition, the feasibility of CXCL10 for the diagnosis and treatment of leprosy was analyzed by a large number of literatures, and their own views were given objectively. The results presented in this systematic review supports the importance of CXCL10 in leprosy diagnosis, particularly in leprosy responses and in tracking the eûcacy of MDT therapy. Using CXCL10 in clinical settings might help with leprosy early diagnosis. In addition, I highly apprecaite the interesting literature survey methods. Some suggestions and comments should be considered and the manuscript should be significantly improved upon before acceptance.

1. Please clearly mention the full spelling and defination of CXCL10 in Abstract and Introduction, respectively. Then it can be used in the following text. And the abbreviations such as BT, BB, BL-LL and BB, BL-LL used in article 92 are not well explained, which makes reading confusing.
2. In this paper, some researches and possible inferences about the mechanism of action of CXCL10 on leprosy are listed in detail in some literatures, which can give readers a very objective understanding of the latest progress of the current research on CXCL10. However, the authors' view is too objective and lacks bias. Some thoughts and discussions on the mechanism of action in different literatures can be added.
3. In the discussion of the results in lines 39 to 51, the paper mentioned some literatures on CXCL's positive effect on the diagnosis and treatment of leprosy, but it did not specify which literatures these views came from, which should be marked.
4. This paper finds that CXCL10 may be very useful in the early diagnosis of leprosy, and the author rationally points out that there may be errors caused by other variables, which is very valuable for further thinking about how to exclude these variables.
5. The logic of the paragraphs starting from the 95th line of the paper is somewhat confused, and the meaning of the paper can only be barely understood. When writing, the authors should pay more attention to the connection between evidence and evidence, and add conjunctions such as Furthermore, Besides, otherwise, which will make the paper seem scattered.
6. In the paragraph of Quality Assessment, the paper mentions the method of Newcastle-Ottawa Scale (NOS) to assess the quality of the selected studies. The explanation is not detailed enough and the process is not given. It would be more convincing if the authors add some detailed data.
7. RoBANS to assess the risk of bias in the selected studies is adopted in the paragraph of Risk Bias Assessment, which comprehensively considers the influence of various factors, but it is still somewhat brief on the whole. The authors can add an explanation of the risks that may arise from different elements, and add some text to the corresponding Figures.
8. In this section of the paper on CXCL10 in Leprosy Reaction, the listed literature lacks connections. It is best to summarize the similarities and differences, so as to make the logicality more smooth.
9. Please cite relevant illustrations in the text to make the ideas more vivid and reliable. Of course, please mention with the consent of other authors of cited references.
10. This manuscript mainly introduceds the detection of CXCL10 by UCP-LFA technology as a biomarker for leprosy, and the comparison with other biomarkers increased the persuasibility of the viewpoint in this paper. The authors may also consider searching some literature on other detection technologies, such as the detection technology of molecular imprinting technology for CXCL10 biomarker. The diagnostic capability of CXCL10 for leprosy is more convincing.
11. In this paper, the advantages of CXCL10 as a diagnostic tool for leprosy are briefly introduced, which can be expanded into more abundant contents, such as cost saving, drug compliance, and negative diagnostic rate.
12. Several recently recent work can be cited to strengthen the review-topic’s importance and improve the manuscirpt organization. For example,

-Mass Spectrometry-based Proteomics and Glycoproteomics in COVID-19 Biomarkers Identification: A Mini-review, J. Anal. Test. 5, 298-313 (2021). https://doi.org/10.1007/s41664-021-00197-6

- Molecularly Imprinted Ratiometric Fluorescence Nanosensors, Langmuir, 2022, 38, 13305−13312. https://doi.org/10.1021/acs.langmuir.2c01925

- CXCL10 as a biomarker of interstitial lung disease in patients with rheumatoid arthritis. Reumatologia clinica, 2024, 20, 1, 1-7, DOI 10.1016/j.reumae.2023.12.005
13. The manuscript is generally well written, but soem descriptions and format still need to be revised and optimized. Please carefully check and improve the total manuscript.

---

## Round 0.2 · accepted · Accept

All the concerns of the reviewers were addressed, and the manuscript was revised accordingly. Therefore, the revised version of the manuscript is acceptable now.

Reviewer 2 ·

Basic reporting

It is OK

Experimental design

It is OK

Validity of the findings

It is OK

Additional comments

No